# Thrombin Preconditioning of Extracellular Vesicles Derived from Mesenchymal Stem Cells Accelerates Cutaneous Wound Healing by Boosting Their Biogenesis and Enriching Cargo Content

**DOI:** 10.3390/jcm8040533

**Published:** 2019-04-18

**Authors:** Dong Kyung Sung, Yun Sil Chang, Se In Sung, So Yoon Ahn, Won Soon Park

**Affiliations:** 1Department of Pediatrics, Samsung Medical Center, Sungkyunkwan University School of Medicine, Seoul 06351, Korea; dbible@skku.edu (D.K.S.); cys.chang@samsung.com (Y.S.C.); himmel7@skku.edu (S.I.S.); yoon.ahn.neo@gmail.com (S.Y.A.); 2Stem Cell and Regenerative Medicine Institute, Samsung Medical Center, Samsung Biomedical Research Institute, Seoul 06351, Korea; 3Department of Health Sciences and Technology, SAIHST, Sungkyunkwan University, Seoul 06351, Korea

**Keywords:** stem cell, extracellular vesicle, production, microenvironment mimetic

## Abstract

The aim of this study was to determine the optimal preconditioning regimen for the wound healing therapeutic efficacy of mesenchymal stem cell (MSC)-derived extracellular vesicles (EVs). To this end, we compared various preconditioning regimens for both the quantitative and qualitative production of MSC-derived EVs, and their therapeutic efficacy for proangiogenic activity in vitro and cutaneous wound healing in vivo. After preconditioning with thrombin (40 U), H_2_O_2_ (50 μM), lipopolysaccharide (1 μg/mL), or hypoxia (10% O_2_), EV secretion was assessed quantitatively by measuring production per cell and protein quantification, and qualitatively by measuring a proteome profiler and an enzyme-linked immunosorbent assay (ELISA) contained within EVs. The therapeutic efficacy of EVs was assessed in vitro by proliferation, migration and tube formation assays of human umbilical cord blood endothelial cells (HUVECs), and in vivo by quantification of cutaneous wound healing. Thrombin preconditioning optimally boosted EV production and enriched various growth factors including vascular endothelial growth factor and angiogenin contained within EVs compared to other preconditioning regimens. Thrombin preconditioning optimally enhanced proliferation, the migration and tube formation of HUVECs in vitro via *p*ERK1/2 and *p*AKT signaling pathways, and cutaneous wound healing in vivo compared to other preconditioning regimens. Thrombin preconditioning exhibited optimal therapeutic efficacy compared with other preconditioning regimens in promoting proangiogenic activity in vitro and in enhancing cutaneous wound healing in vivo. These preconditioning regimen-dependent variations in therapeutic efficacy might be mediated by boosting EV production and enriching their cargo content.

## 1. Introduction

Cutaneous wound healing is a complex but well-orchestrated sequential process including coagulation and hemostasis, cell migration, inflammation, angiogenesis, proliferation and remodeling [1,2]. However, when this orderly healing process is disrupted by factors such as severe cutaneous burns, repeated trauma, infection, and/or ischemia, particularly in medical conditions including diabetes mellitus, the wounds might fail to heal properly, last for months or even years, and could thus be defined as chronic non-healing wounds [3,4]. Currently, while many patients suffer from non-healing wounds due to diabetes mellitus [5] and cutaneous burns [6], about 50% of these chronic non-healing wounds do not respond to current wound healing treatments [7] and could ultimately lead to a major disability and even death [6]. Therefore, developing new and effective wound healing therapies, especially against chronic non-healing cutaneous wounds, is urgent.

Recently, stem cell therapy, especially using mesenchymal stem cells (MSCs), has emerged as a promising new and effective therapeutic strategy for treating chronic non-healing wounds, and this protective effect was predominantly mediated by paracrine rather than direct regenerative mechanisms [8,9]. Recent studies have demonstrated that the beneficial effects of MSCs for wound healing were only some mediated by the secretion of extracellular vesicles (EVs), nuclear membrane vesicles 40–100 nm in diameter and containing numerous proteins, lipids, mRNAs, and regulatory miRNAs [9,10]. Since this cell-free therapy can bypass concerns associated with viable MSC transplantation, the use of MSC-derived EVs might be a promising new, safe, and effective therapeutic modality for cutaneous wound healing.

Preconditioning MSCs with various stressors such as thrombin, hypoxia, lipopolysaccharide (LPS), and hydrogen peroxide (H_2_O_2_) has been known to stimulate the biogenesis of MSC-derived exosomes, modify their cargo content, and enhance therapeutic potential of MSC-derived EVs [11,12,13,14,15,16]. However, to the best of our knowledge, the therapeutic efficacies of various preconditioning regimens for enhancing cutaneous wound healing have not yet been directly compared. The aim of the present study was thus to determine the optimal preconditioning regimen for accelerating cutaneous wound healing. To this end, the therapeutic efficacies of preconditioning with thrombin, hypoxia, LPS, and H_2_O_2_ in protecting against both in vitro and in vivo cutaneous wounds were directly compared. We also evaluated whether preconditioning regimen-dependent variations in therapeutic efficacies were associated with or mediated by boosting EV production and enriching EV content by measuring the amount of EV secretion quantitatively and by measuring a proteome profiler and an enzyme-linked immunosorbent assay (ELISA) contained within EVs qualitatively.

## 2. Experimental Section

### 2.1. Mesenchymal Stem Cells

Human umbilical cord blood-derived MSCs (UCB-MSCs) from a single donor at passage 6 were manufactured in strict compliance with good manufacturing practice. UCB-MSCs were obtained from Medipost Co., Ltd. (Seoul, Korea) [17,18]. Characterization of MSCs was previously described [19]. The differentiation potentials [17,20] and immunophenotypic results [20] of this process have also been previously described. In 11 passages of UCB-MSCs, no changes in the karyotype were observed [21]. MSCs from a single donor at passage 6 were used in this study.

### 2.2. Preconditioning of MSCs

UCB-MSCs were cultured in α-MEM (Gibco, Grand Island, NY, USA) supplemented with 10% vol/vol fetal bovine serum (Gibco), 100 units/mL penicillin, and 100 μg/mL streptomycin (Invitrogen, Carlsbad, CA) under standard culture conditions. At approximately 90% confluency, the cells were washed with PBS three times to avoid contamination by FBS-derived extracellular vesicles. Cells were exposed to fresh serum free α-MEM supplemented with thrombin (5 U, 10 U, 20 U, 40 U), LPS (100 ng, 1 μg, 10 μg, 100 μg), and H_2_O_2_ (10 μM, 50 μM, 100 μM, 200 μM) for 6 h. To expose hypoxic preconditions, the cells were moved from standard culture conditions to 10%, 5%, and 1% achieved using oxygen-controlled incubator Galaxy 48R for 6 h. To quantify the amount of EV production by single cells after collecting the conditioned media, cells were counted with the LUNA-FL (Logos Biosystems, Anyang-si, Korea) according to the manufacturer’s protocol.

### 2.3. Isolation of EVs

After preconditioning, the conditioned media were centrifuged at 3000 rpm. for 30 min at 4 °C (Eppendorf, Hamburg, Germany) to remove cellular debris, followed by centrifugation at 100,000 rpm for 120 min at 4 °C (Beckman, Brea, CA, USA) to sediment the EVs. This pellet was washed twice, re-suspended in sterile PBS, and then stored at −80 °C until further use.

### 2.4. Quantification of EV Production

The total EV protein content was quantified by measuring the protein concentration using the Bradford assay. EV distribution was analyzed by measuring the rate of Brownian motion using a NanoSight (NanoSight NS300, Malvern, Worcestershire, UK), which is equipped with fast video capture and particle-tracking software. The obtained EVs were resuspended in PBS (500 μL, 1 mg/mL total protein) and characterized for size and polydispersity. To quantify the amount of EV production by single cell after collecting the conditioned media, cells were counted with the LUNA-FL according to the manufacturer’s protocol. The number of EVs produced by a single cell was calculated by dividing the measured number of cells by the total number of EVs. Dynamic light scattering and zeta potential determinations were performed with a Zetasizer nanoseries instrument (Malvern Instruments Ltd., Worcestershire, UK, λ = 532 nm laser wavelength) to determine the EV size. The EV size data refers to the scattering intensity distribution (z-average).

### 2.5. Early Endosome Labeling

Early endosomes were labelled using CellLight Early Endosomes-GFP (Invitrogen, Carlsbad, CA, USA). Fifteen thousand cells were seeded in 8-well microscopy slides and incubated for 24 h before replacing the medium with CellLight Early Endosomes-GFP at a concentration of 40 virus particles per cell.

### 2.6. Angiogenesis Antibody Array Assay

The relative expression of 55 angiogenesis-related proteins was determined in each lysate using a Proteome Profiler™ human angiogenesis array kit (R&D Systems, Abingdon, UK) following the manufacturer’s protocol. Briefly, following blocking of the nitrocellulose membrane spotted with antibodies against angiogenesis-related proteins, 100 μg of extracellular vesicle lysates were mixed with a cocktail of biotinylated detection antibodies (15 μL), and this mixture was added to the membrane before incubating overnight at 4 °C. Streptavidin-horseradish peroxidase (HRP, 1.5 mL) was then added to the membrane, and the resulting mixture was incubated for 30 min before chemiluminescence detection reagents were added in equal volumes for 1 min. Residual detection reagent was carefully blotted off the membrane before it was covered in plastic wrap and exposed to X-ray film for 5 or 10 min. Intensity of protein expression was quantified with Image J software (National Institutes of Health, Bethesda, MD, USA), which indicated that the total mean intensity was equal to the area multiplied by the mean intensity.

### 2.7. Bioplex Assay

The Fluorokine^®^ MAP, Human Angiogenesis Custom Premix Kit A (R&D Systems, Minneapolis, MN, USA), was used to quantify angiogenin, angiopoietin-1, VEGF, and HGF in EVs, according to the manufacturer’s instructions. The cytokine level of the wound tissues was analyzed by ELISA. Rats were sacrificed four or eight days post-treatment for cytokine analysis of wounds. The homogenate of dissected wound tissues was added to the well of an ELISA kit containing 0.1 mL of the lysis buffer. Following the Bradford method, preparations were normalized for protein content and 10 μg were loaded for each ELISA well. The Fluorokine^®^ MAP, Rat cytokine Custom Premix Kit A (R&D Systems, Minneapolis, MN, USA), was used to quantify TNF-a and IL-6 in wound tissue according to the manufacturer’s instructions.

### 2.8. Bovine Thrombin Residual Measurements

A bovine thrombin ELISA kit (San Diego, CA, USA) was used to quantify thrombin in EVs, according to the manufacturer’s instructions. Following the Bradford method, preparations were normalized for protein content and 20 μg were loaded for each ELISA well. The absorbance of each well was determined with the Bio-Rad microplate reader (xMark™Spectrophotometer, Bio-Rad Laboratories, Inc., Hercules, CA, USA) at 450 nm.

### 2.9. Western Blot Analysis

EV preparations were lysed by adding an equal volume of radioimmunoprecipitation assay (RIPA) buffer (Sigma–Aldrich, St. Louis, MO, USA). Following the Bradford method, preparations were normalized for protein content, and 10 μg were loaded in each. Loading buffer containing β-mercaptoethanol was added, and the EV preparations were boiled for 10 min, separated in a 12% SDS-polyacrylamide gel gradient (SDS-PAGE), and electrophoretically transferred to nitrocellulose membranes. Membranes were blocked in 5% BSA in phosphate-buffered saline-Tween-20 (0.5%) at RT and incubated for 1 h with the primary antibodies at RT. Membranes were then washed with 1x PBST. The secondary antibodies used were the anti-mouse or -rabbit IgG conjugated HRP (1:2000) (Dako, Ely, UK). Secondary antibodies were incubated for 1 h with agitation at RT. Following PBST washes, protein bands were detected using the chemiluminescence reagent ECL Select (GE Healthcare Life SciencesTM, Waukesha, WI, USA), and images were acquired with X-ray films.

### 2.10. In Vitro Matrigel Tube Formation Assay

HUVECs (5 × 10^4^ cells/mL) were cultured on growth factor-reduced Matrigel (BD-Biosciences CA USA)-coated plates for 17 h, in a medium containing 10% plasma or Ham F12-K medium alone. Tube formation was examined by phase-contrast microscopy 24 h later. Each condition in each experiment was assessed in duplicate, and tube length was measured as the mean of the summed length of capillary-like structures in two wells, in per high-power fields (HPFs, 100×) per well. Three randomly selected HPFs (100×) were analyzed in each section. Six to nine experiments were performed under each condition. A semi-quantitative measurement of cord formation in the entire Matrigel culture was performed and expressed as a tube-formation index [22].

### 2.11. In Vitro Proliferation and Viability Assays

Cell viability was evaluated via the MTT assay. UCB-MSCs, human dermal fibroblast, or HUVECs (1 × 104 cells/well) were seeded on 96-well flat-bottomed plates and incubated with EVs for 24 h at 37 °C; the MTT assay reagent (Promega Corporation, Madison, WI, USA) was added to the wells for 3 h at 37 °C. Viability was evaluated by measuring the absorbance at a 490 nm wavelength with a 96-well ELISA plate reader (xMark™Spectrophotometer, Bio-Rad Laboratories, Inc., Hercules, CA, USA) in at least six wells per experiment, and there were eight experiments per condition.

### 2.12. In Vitro Scratch-Wound-Closure Assays

Human dermal fibroblast cells were seeded into 6-well dishes at a density of 0.5 × 10^5^ cells per well. The cells were grown to confluence in 6-well plates. After a 16 h incubation serum free condition, the cells then scored with a sterile pipette tip to leave a scratch of approximately 0.4–0.5 mm in width. The removed medium was replaced in DMEM 0.5% FBS with (i) an EV-free culture medium (NC), (ii) a culture medium containing thrombin-treated EVs, (iii) a culture medium containing 10% O_2_-treated EVs, (iv) a culture medium containing LPS (1 μg/mL) -treated EVs, or (v) a culture medium containing H_2_O_2_ (50 μM)—treated EVs. Wound-closure effects were observed at least 48 h later. Percent of wound closure was calculated as [(*Area* 0 *h* − *Area* 48 *h*)/*Area* 0 *h*] × 100.

### 2.13. Animal Model

All animal procedures were reviewed and approved by the Institutional Animal Care and Use Committee of Samsung Biomedical Research Institute, Seoul, Korea (20150520002). These procedures were also performed in accordance with our institutional guidelines, in addition to the National Institutes of Health Guidelines for Laboratory Animal Care. All animal procedures were performed in an Association for Assessment and Accreditation of Laboratory Animal Care accredited specific pathogen-free facility. Eight-week-old male SD rats, purchased from Orient Co. (Seoul, Korea), were housed in individual cages with free access to water and laboratory chow. All rats were anesthetized during the operation using intraperitoneal injections of ketamine (100 mg/kg, Yuhan Co. Seoul, Korea) and xylazine (1.2 mg/kg, Bayer Healthcare, Kyunggido, Anseong, Korea). In this study, we used a humane endpoint as the earliest indicator of pain or distress; to avoid or limit pain and distress, euthanasia was performed. For the humane endpoint, an operationally defined scoring system was approved by Institutional Animal Care and Use Committee. A total score of five or more or a score of three in any single category was arbitrarily defined as the humane endpoint. The dorsal skin was shaved and cleaned with alcohol. A full-thickness wound (8-mm) was made on the dorsal skin of the SD-rat using a punch biopsy. The wound was not covered. Four animals were used per group. The wounds were photographed using a digital camera two, four, six, and eight days after surgery, and the wound area was measured by tracing the wound margin and performing the calculation using ImageJ (NIH, Bethesda, MD, USA). The wound area was analyzed by calculating the percentage of the current wound with respect to the original wound area. The wound was considered to be completely closed when the wound area was grossly equal to zero.

### 2.14. Transmission Electron Microscopy (TEM)

UCB-MSCs were fixed with 4% paraformaldehyde and 1% glutaraldehyde in 0.1 M sodium cacodylate buffer (pH 7.2) (Electron Microscopy Sciences, Hatfield, PA, USA) for 3 h at room temperature, washed with cacodylate buffer, postfixed in 1% osmium tetroxide, progressively dehydrated in a graded ethanol series (50%–100%), and embedded in Epon. Thin (1 mm) and ultrathin (70–80 nm) sections were cut from the polymer with a Reichert (Depew, NY, USA) Ultracut S microtome, placed on copper grids, and briefly stained with uranyl acetate (UA) and lead citrate.

Five microliter EVs were fixed with 2% glutaraldehyde, loaded on 200-mesh formvar/carboncoated electron microscopy grids (Electron Microscopy Sciences, PA, USA), and incubated for 10 min. They were then washed with filtered distilled water and stained using 2% UA in water for 1 min. Transmission electron microscopy images were obtained with an FEI (Hillsboro, OR, USA) Tecnai Spirit G2 transmission electron microscope operating at 120 kV.

### 2.15. Scanning electron microscopy (SEM)

Isolated EVs were fixed with 2.5% glutaraldehyde and loaded on the polycarbonate membrane. Membrane was washed once PBS, water and dehydrated with acetone. The acetone was subsequently removed using critical point drying with liquid carbon dioxide. Samples were mounted on aluminum stubs with carbon tape mounted on a SEM stub after sputter-coating with 3–5 nm platinum the samples were examined with a SEM (Zeiss Auriga Workstation, Oberkochen, Germany).

### 2.16. Statistical Analyses

All quantitative results were obtained from triplicate samples. Data were expressed as a mean ± SD, and the statistical analyses were carried out using two-sample t-tests to compare two groups of samples and a One-way Analysis of Variance (ANOVA) for the three groups. A value of *p* < 0.05 was considered to be statistically significant.

## 3. Results

### 3.1. Isolation and Characterization of MSC-Derived EVs

We isolated the EVs from conditioned media using ultracentrifuge methods and characterized the MSC-derived exosomes by scanning electron microscopy (Figure 1A) and transmission electron microscopy (Figure 1B). TEM data show that EVs were released from the cell membrane. (Figure 1C) EV size distribution was quantified by a Zetasizer. (Figure 1D). Particle size distribution displayed about an 85% range from 30 to 100 nm. We next validated an exosomal protein marker such as CD63, CD9, and CD81 via Western blotting. The presence of CD63, CD9, and CD81 supported the existence of an endosomal origin of the secreted vesicles, consistent with the exosomal nature of these vesicles. (Figure 1E)

### 3.2. Determining Optimal Preconditioning Regimen for Optimal EV production

EV production, cytotoxicity, and apoptosis were analyzed through protein quantification (Figure 2A), nanoparticle tracking analysis (Figure 2B), an MTT assay (Figure 2C), Terminal deoxynucleotidyl transferase dUTP nick end labeling (TUNEL) staining (Figure 2D), and caspase 3 Western blotting (Figure 2E) with various concentrations of preconditioning regimens. Thrombin preconditioning from 1 to 40 U showed dose-dependent EV production without cytotoxicity nor apoptosis. LPS from 100 ng to 100 μg showed a significant, but not a dose-dependent, increase in EV production compared with the naïve MSCs without cytotoxicity or apoptosis. H_2_O_2_ from 10 to 200 μM showed a significant, but not a dose-dependent, increase in EV production compared with the naïve MSCs. When the UCB-MSC-treated H_2_O_2_ with more than 50 uM, apoptosis and caspase-3 expression were observed. Hypoxia from 10 to 1% showed a significant, but not a dose-dependent, increase in EV production compared with the naïve UCB-MSCs culture condition. When UCB-MSCs cultivated oxygen conditions of less than 10%, apoptosis and caspase-3 expression were observed. Based on these results, an optimal preconditioning dose of 40 U thrombin, 1 μg of LPS, 50 μM of H_2_O_2_, and 10% hypoxia were used in this study.

### 3.3. Effects of Preconditioning on EV Production and Characteristics

Transmission electronic microscopy showed that more EVs were observed near the thrombin-treated UCB-MSC membrane. Although the endoplasmic reticulum was well observed in the UCB-MSCs, the EV formation in the endoplasmic reticulum was more observed in the thrombin-treated group than others (Figure 3A). Early endosomes were labelled using CellLight Early Endosomes-GFP with green fluorescence. More green fluorescence particles were observed in the thrombin treatment group within in the UCB-MSCs (Figure 3B).

Not all the precondition regimens affected the peak size distribution (50–70 nm in diameter) of the isolated spheroidal EVs (Figure 3C,D). While all the preconditioning regimens increased EV production compared with the naïve MSCs, thrombin preconditioning enhanced EV production more compared with the other preconditioning regimens (Figure 3E). On the other hand, the EVs were positive for exosome-specific CD9, CD63, and CD81 markers. Mitochondrial Cytochrome C was observed in the H_2_O_2_- and hypoxia-preconditioned EVs but not in the naïve EVs or the thrombin- or LPS-preconditioned EVs. GM130 and fibrillarin were not observed in all experimental groups (Figure 3F).

### 3.4. Effects of Preconditioning on Protein Cargo in Shedding EVs

We assayed a Proteome Profiler Human Angiogenesis Array and ELISA to determine whether changes in the various UCB-MSC culture conditions caused changes in the angiogenesis factors in EVs isolated from UCB-MSCs. Figure 4 demonstrates the preconditioning regimen-dependent variation in the protein cargo content with the Proteome Profiler Human Angiogenesis Array and the ELISA assay. All the preconditioning regimens except LPS preconditioning increased the angiogenic growth factors such as angiogenin, angiopoietin-1, HGF, and Vascular endothelial growth factor (VEGF) within the EVs compared with the naïve MSC EVs (Figure 4A). Angiogenin, angiopoietin-1, HGF, and VEGF more than tripled compared to naïve controls, when confirmed the protein level using the ELISA method (Figure 4B). Thrombin preconditioning optimally enhanced the angiogenic protein cargo contained within the EVs compared with the preconditioning regimens with H_2_O_2_ and hypoxia.

### 3.5. Effects of Preconditioning on Proangiogenic Effects and ERK Phosphorylation of EVs in Vitro

EVs isolated thrombin-treated UCB-MSCs were measured using ELISA to determine how much thrombin was included, since thrombin is a known mitogen factor that is a potent promoter of angiogenesis [23,24]. There were no differences between group of negative control, naïve EVs, and thrombin EVs, which suggest no contamination of thrombin in thrombin EVs (Appendix A). All preconditioning regimens except LPS promoted HUVEC proliferation significantly more (Figure 5A), although fasting serum inhibit cell proliferation. Figure 5C shows that the HUVECs cultured in EVs derived from thrombin-treated UCB-MSC-conditioned media markedly produced capillary-like tubes compared with HUVECs cultured in EVs derived from LPS-, H_2_O_2_-, and hypoxia-treated UCB-MSC-conditioned media. Quantitative analysis shows that thrombin-treated EVs are statistically superior to LPS-, H_2_O_2_-, and hypoxia-treated EVs (Figure 5B). These results indicated that thrombin preconditioning optimally improved the proangiogenic effects of EVs compared with preconditioning regimens with H_2_O_2_ and hypoxia. The activation of MAPK and Akt signaling pathways is known to be involved in the proliferation of HUVECs [25,26]. Thus, we investigated the effects of EVs derived from UCB-MSCs on the phosphorylation of Akt and ERK1/2 in HUVECs. The *p*ERK1/2 and *p*AKT of EVs were significantly increased with all preconditioning regimens, except for that with LPS preconditioning, compared with the naïve MSC-derived EVs, and optimally increased with thrombin preconditioning compared with preconditioning regimens with H_2_O_2_ and hypoxia (Figure 5D).

### 3.6. Effects of Preconditioning on Human Dermal Fibroblast Proliferation and the ERK Phosphorylation of EVs in Vitro

All preconditioning regimens except for LPS preconditioning significantly promoted human dermal fibroblast (hDF) proliferation (Figure 6A). hDF growth and migration rates were calculated by measuring cell coverage into the scratched space at defined time points at 0 h and 48 h. Images of the scratch wounds on hDF monolayers were captured 0 and 48 h after scratching to show that hDF proliferation into the scratch wound area was accelerated in the thrombin EV-treated group compared with other EV-treated groups (Figure 6C). Quantitative analysis confirmed that thrombin-treated EVs were superior to naïve EVs and to LPS-, H_2_O_2_-, and hypoxia-treated EVs. (Figure 6B) respectively. The activation of MAPK and Akt signaling pathways is known to be involved in the migration and proliferation of skin fibroblasts [27]. Thus, we investigated the effects of EVs derived from UCB-MSCs on the phosphorylation of Akt and ERK1/2. The blot intensity was higher in the thrombin-treated EV group than in the H2O2-, LPS-, and hypoxia-treated EV groups (Figure 6D).

### 3.7. Effects of Preconditioning on the Cutaneous Wound Healing of EVs

With EV treatment obtained from the same number (5 × 10^5^) of MSCs with various preconditioning regimens, only EVs obtained from thrombin preconditioning significantly improved the wound closure rate compared with the control group. Figure 7A shows photographs of wound healing in rats subjected to skin excision wounds after zero, two, four, six, and eight days. The results show that thrombin-stimulated EVs significantly improved the rate of wound healing compared with other treatment groups after two, four, six, and eight days (Figure 7B). TNF-α and IL-6 levels measured using the ELISA method were statistically lower in the thrombin-treated EVs, compared to other EV groups on the fourth and eighth days.

With the same amount of EV protein (20 μg/10 uL) treatment, a significantly improved cutaneous wound closure rate and less inflammatory cytokines such as TNF-α and IL-6 were observed in both the naïve and the preconditioned EVs compared with the saline-treated group. The thrombin-, H_2_O_2_-, LPS-, and hypoxia-treated EV groups had an improved wound closure rate compared with the naïve EV-treated groups. However, thrombin preconditioning more improved the cutaneous wound closure rate compared with the naïve and the preconditioning regimens with LPS, H_2_O_2_ and hypoxia (Figure 8A,B). The thrombin-, H_2_O_2_-, LPS-, and hypoxia-treated EV groups had reduced TNF-α and IL-6 levels compared with the saline-treated control group. The thrombin-treated group had significantly lower level of cytokines compared to the naïve group and the groups treated with LPS, H2O2, and hypoxia. (Figure 8C).

## 4. Discussion

Compared to their parent MSCs, EVs have a superior safety profile and can be safely stored without losing function, and are thus more suitable as an “off the shelf” drug [28]. Despite these promising results, the small amount of EVs constitutively secreted by MSCs is a challenge in studying EVs. While the inducible secretion of EVs can be enhanced by preconditioning with stress stimuli including thrombin, hypoxia, H_2_O_2_, and LPS, the efficacy of various preconditioning regimens for boosting the secretion of EVs has not yet been directly compared. Therefore, an optimal preconditioning regimen protocol for enhancing the secretion of MSC EVs is lacking. In the present study, while all the preconditioning regimens with thrombin, hypoxia, H_2_O_2_, and LPS significantly enhanced EV biogenesis, evidenced by more ILVs, MVBs, and secretion compared with the naïve MSCs, thrombin preconditioning with up to 40 U was more than fourfold effective compared with other preconditioning regimens in terms of stimulating the inducible biogenesis and secretion of MSC-derived EVs independent of cytotoxicity and apoptosis. These findings suggest that thrombin preconditioning can easily upscale the production of EVs by more than four times compared with other preconditioning regimens without any concern of cytotoxicity or apoptosis.

Besides enhancing the production of EVs, preconditioning regimens could also affect the cargo composition of EVs [29,30,31]. In this study, while all the preconditioning regimens except LPS preconditioning increased the angiogenic growth factors such as angiogenin, angiopoietin-1, HGF, and VEGF within the EVs compared with the naïve MSC-derived EVs, thrombin preconditioning optimally enhanced the angiogenic protein cargo content within the EVs compared with the preconditioning regimens with H_2_O_2_ and hypoxia. Overall, these findings suggest that that thrombin preconditioning is optimally effective for boosting EV biogenesis, secretion, and cargo loading with proangiogenic factors.

In the present study, in strict accordance with the extent of enriched EV cargo content of proangiogenic growth factors, thrombin preconditioning optimally enhanced the proangiogenic activity of EVs in vitro including the proliferation, migration, and tube formation of HUVECs compared with preconditioning regimens with H_2_O_2_ and hypoxia. In line with our data, increased VEGF and miR-126 levels in exosomes released from MSCs by nitric oxide stimulation more effectively promoted their proangiogenic processes [31]. Moreover, in our previous study [32], when we blocked the VEGF within the EVs using VEGF siRNA-transfected MSCs, the protective effects of EVs against neonatal hyperoxic lung injuries, such as the significant attenuation of inflammatory responses and the ensuing impaired alveolarization and angiogenesis were abolished. Collectively, these findings suggest that proangiogenic growth factors within EVs dose-dependently determine their proangiogenic activity in vitro including the proliferation, migration, and tube formation of HUVECs [31,33]. For in vivo cutaneous wound healing, the significant attenuation of inflammatory responses and an improved closure rate were observed only with preconditioned thrombin but not with other preconditioned EVs obtained from the same amount of MSCs. Furthermore, even though an equal dose was administered in all EV treatment groups, both naïve and preconditioned EVs showed significantly increased anti-inflammatory effects and improved wound closure rates compared with the control group, and thrombin-preconditioned EVs optimally attenuated the inflammatory cytokines and improved wound closure rates compared with the other preconditioning regimens. Overall, these findings suggest that both the quantity and quality of EVs are equally important for determining the therapeutic efficacy of EV treatment.

In summary, thrombin preconditioning optimally boosted MSC-derived EV production, and enriched their cargo content with proangiogenic growth factors compared with preconditioning regimens with H_2_O_2_. Furthermore, thrombin preconditioned EVs were optimally effective for promoting cell proliferation, migration, and tube formation of HUVECs in vitro, and accelerating the cutaneous wound healing in vivo. This simple, safe, and effective thrombin preconditioning regimen for better EV treatment could be easily translated into clinical use.

## 5. Conclusions

Thrombin preconditioning exhibited therapeutic in promoting proangiogenic activity in vitro and in enhancing cutaneous wound healing in vivo. These preconditioning regimen-dependent variations in therapeutic efficacy might be mediated by boosting EV production and enriching their cargo content.

## Figures and Tables

**Figure 1 jcm-08-00533-f001:**
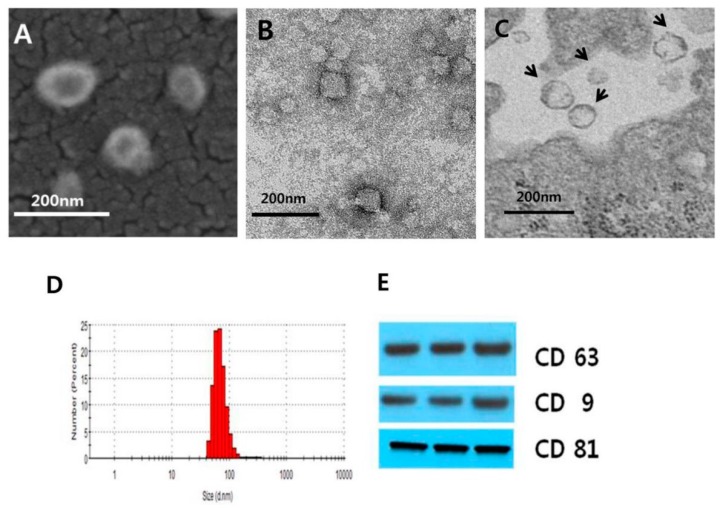
Characterization of extracellular vesicles (EVs). (**A**) Scanning electron microscopic image of EVs loaded on a polycarbonate membrane. EVs were isolated from conditioned media of cultures for human umbilical cord blood-derived mesenchymal stem cell (MSC) using ultra-centrifugation. (**B**) Transmission electron microscope (TEM) photograph of EVs. EVs were placed on copper grids and stained with uranyl acetate with showing round shape. (**C**) TEM image of EVs being secreted from an MSC membrane. (**D**) Particle size distribution of EVs using Zetasizer analysis. The Y-axis represents the number of EVs and the X-axis represents the size of EVs analyzed by the Zetasizer showing a main peak around 60–80 nm. (**E**) Western blots for three consecutively separated EVs, indicating exosome marker proteins of CD63, CD9, and CD81, respectively.

**Figure 2 jcm-08-00533-f002:**
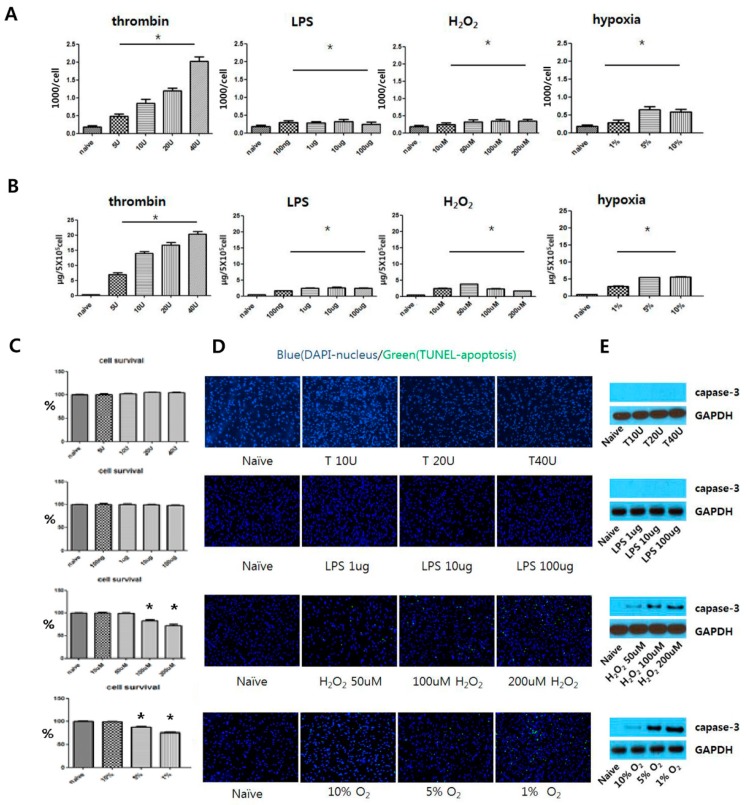
Effects of various preconditioning for umbilical cord blood-derived mesenchymal stem cells (UCB-MSCs) on exracellular vesicle (EV) production, cell viability, and cell death. MSCs were preconditioned by treatment of thrombin (5 U, 10 U, 20 U, and 40 U), LPS (100 ng, 1 μg, 10 μg, and 100 μg), H_2_O_2_ (10 μM, 50 μM, 100 μM, and 200 μM), and hyoxia (10% O_2_, 5% O_2_, 1% O_2_) for 6 h, respectively. (**A**) Representative nanoparticle tracking (NTA) analysis for isolated EVs expressed with total counts of EVs divided by the total number of MSCs. (**B**) Total protein concentration of isolated EVs from 5 × 10^5^ MSCs measured by the Bradford method. (**C**) Each preconditioning status was processed for 6 h and analyzed by the MTT assay after 24 h. (**D**) Terminal deoxynucleotidyl transferase dUTP nick end labeling (TUNLE) staining for MSCs in each culture condition. TUNEL-positive cells were labeled with fluorescein isothiocyanate (FITC) (green) and the nuclei with DAPI (blue). (**E**) Representative Western blot for caspase-3 expression of MSCs in each preconditioning status. Data are presented as mean ± SEM. Statistical analyses were carried out using a One-way Analysis of Variance (ANOVA). The asterisk (*) indicates the *p* < 0.05 compared with naïve UCB-MSCs.

**Figure 3 jcm-08-00533-f003:**
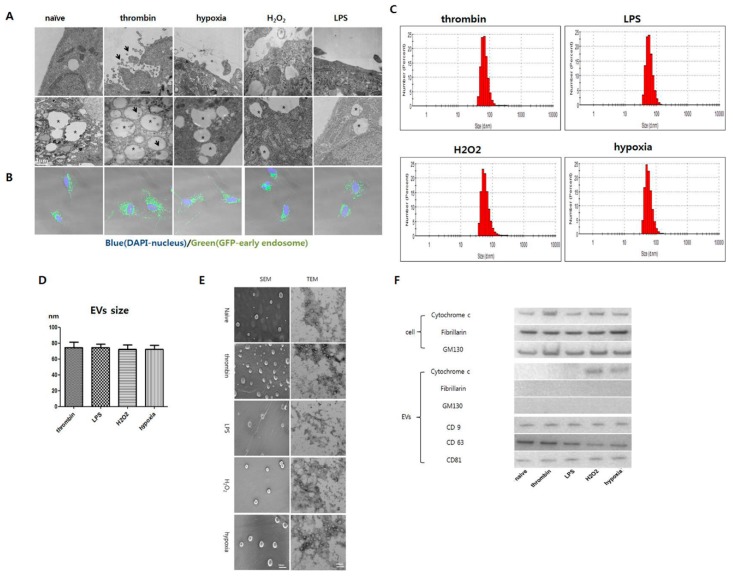
Endosome and characterization EVs derived from MSCs with various preconditioning. MSCs were preconditioned with thrombin (40 U), hypoxia (10% O_2_), LPS (1 μg), or H_2_O_2_ (50 μM) respectively. (**A**) Representative TEM image of MSCs. In the thrombin-preconditioned group, numerous EVs (black arrow) are observed abundantly near the cell membrane but observed sparsely in other groups (upper panel). Although endosomes are well observed in MSCs in all preconditioned groups (asterisk), more EVs are observed (black arrow) in thrombin-treated MSCs than others (lower panel). (**B**) Early endosomes labelled with CellLight Early Endosomes-GFP in MSCs. Endosomes were labeled with GFP (green) and the nuclei with DAPI (blue). (**C**) Particle size distribution of EVs using a Zetasizer. The Y-axis represents the number of EVs and the X-axis represents the size of EVs analyzed by a Zetasizer. (**D**) The average size of isolated EVs from MSCs cultured in various preconditioning status. (*n* = 5). (**E**) Representative SEM (left panel) and TEM (right panel) of isolated EVs from MSCs cultured with various preconditioning statuses. (**F**) Representative Western blot for the organelle marker protein from MSCs and MSC-derived EVs, cytochrome C for the mitochondria marker, fibrillarin for the nucleus marker, GM130 for the Golgi apparatus marker, and CD63 and CD9 for the exosome marker.

**Figure 4 jcm-08-00533-f004:**
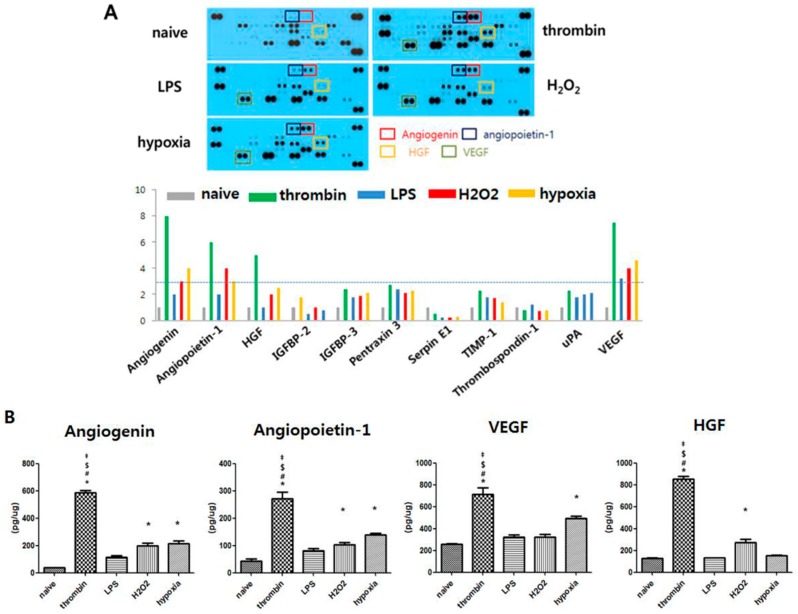
The EVs contain various cytokines and chemokines following stimulation with different wound mimetic environments. EVs were isolated from MSCs preconditioned with thrombin (40 U), LPS (1 μg,) H_2_O_2_ (50 μM), or hypoxia (O_2_ 10%). (**A**) Representative screening X-ray film blot of Proteome Profiler Human Angiogenesis Array (PPHAA) for protein cargos of EVs (upper panel) and their representative diagram for the relative content of each protein expressed by fold change of blot intensity normalized with that of the naïve group. A gray dot line represents a three-fold increase over that of the naïve group. (**B**) Levels of angiogenin, angiopoietin-1, VEGF, and HGF measured by muliplex ELISA in EVs. Statistical analyses were carried out using a One-way Analysis of Variance (ANOVA). The asterisk (*) indicates *p* < 0.05 compared with naive EVs, the number sign (#) indicates *p* < 0.05 compared with the 10% oxygen-treated EV group, the double dagger (‡) indicates *p* < 0.05 compared to the H_2_O_2_-treated EV group, and the dollar sign ($) indicates *p* < 0.05 compared with the LPS-treated EV group (*n* = 5 per each analysis).

**Figure 5 jcm-08-00533-f005:**
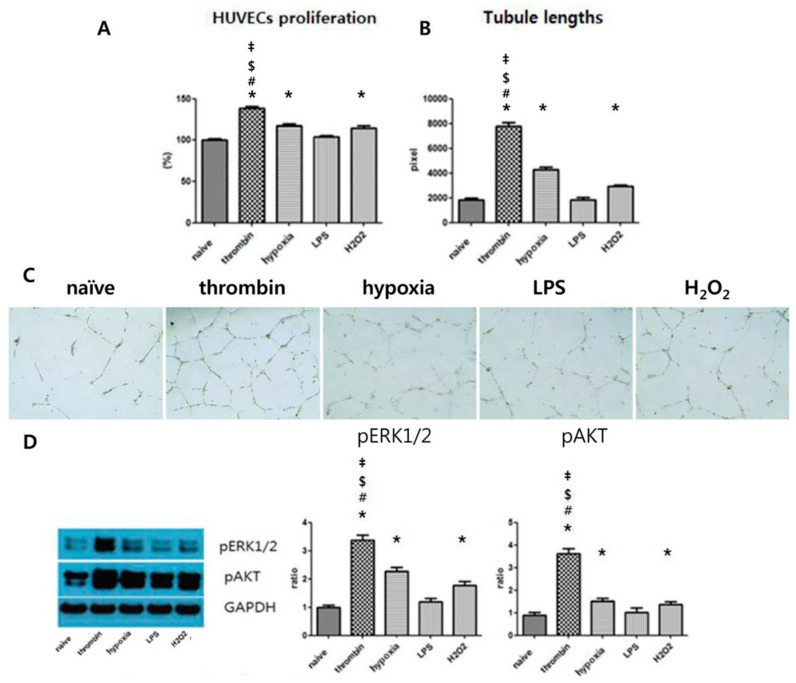
The promotion of proliferation and angiogenesis of HUVECs with stimulation by EVs. (**A**) Proliferation rates of HUVECs following treatment with naïve and thrombin-, 10% O2-, LPS-, and H_2_O_2_-treated EVs for 24 h. HUVECs were cultured on Matrigel with a medium containing naïve MSC-EVs, thrombin-treated (40 U) EVs, 10% O_2_-treated EVs, LPS-treated (1 μg) EVs, or H_2_O_2_-treated (50 μM) EVs. (**B**) Quantitation of pixels and (**C**) representative photomicrographs (×100). (**D**) Lysate from HUVECs treated with EVs for 24 h subjected to Western blot analysis. Cell lysate was analyzed by immunoblotting and normalized to GAPDH, and the ratio of densitometry to GAPDH is shown beneath the blots. Data are presented as mean ± SEM Statistical analyses were carried out using a One-way Analysis of Variance (ANOVA). The asterisk (*) indicates *p* < 0.05 compared with naive EVs, the number sign (#) indicates *p* < 0.05 compared with the 10% oxygen-treated EV group, the double dagger (‡) indicates *p* < 0.05 compared to the H_2_O_2_-treated EV group, and the dollar sign ($) indicates *p* < 0.05 compared with the LPS-treated EV group (*n* = 5 per each analysis).

**Figure 6 jcm-08-00533-f006:**
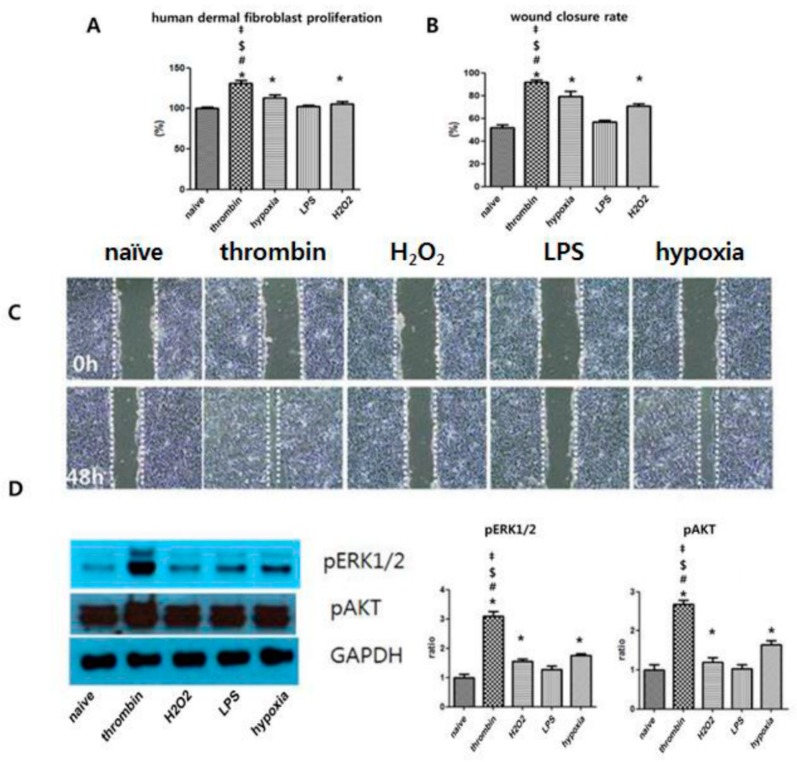
The promotion of proliferation of human dermal fibroblasts with stimulation of EVs. (**A**) Proliferation rates of human fibroblasts following treatment with normal and thrombin-, 10% O_2_-, LPS-, H_2_O_2_-treated EVs for 24 h. Wound closure rate was detected by wound scratch assays as described in Materials and Methods. (**B**) Rates of wound closure were quantified using Image J. (**C**) Representative images of wound scratch assays at the treatment of the normal control and with thrombin, H_2_O_2_, LPS, and hypoxia. (**D**) Lysate from human dermal fibroblast cells treated with EVs for 24 h subjected to Western blot analysis. Cell lysate was analyzed by immunoblotting and normalized to GAPDH, and the ratio of densitometry to GAPDH is shown beneath the blots. Data are presented as mean ± SEM. Statistical analyses were carried out using a One-way Analysis of Variance (ANOVA). The asterisk (*) indicates *p* < 0.05 compared with naive EVs, the number sign (#) indicates *p* < 0.05 compared with the 10% oxygen-treated EV group, the double dagger (‡) indicates *p* < 0.05 compared to the H_2_O_2_-treated EV group, and the dollar sign ($) indicates *p* < 0.05 compared with the LPS-treated EV group (*n* = 5 per each analysis).

**Figure 7 jcm-08-00533-f007:**
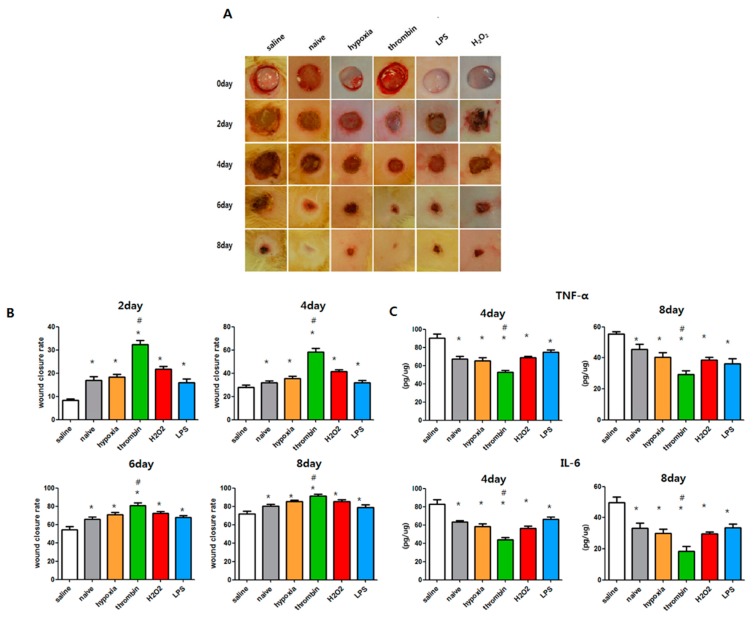
Comparison of wound healing capacity of EVs isolated from the same cell number of MSCs (5 × 10^5^) in vivo. Punch wounds of animals were treated locally with saline, naive EVs, 10% O2 EVs, thrombin (40 U) EVs, LPS (1 μg) EVs, or H2O2 (50 μM) EVs derived from the same number of MSCs. (**A**) Representative serial wound images in vivo and (**B**) the relative wound closure rate (%) after 2, 4, 6, and 8 days of treatment. (**C**) Cytokine analysis for TNF-α and IL-6 β in the homogenate of dissected wound tissues four and eight days after each treatment. Data are presented as mean ± SEM. Statistical analyses were carried out using a One-way Analysis of Variance (ANOVA). The asterisk (*) indicates *p* < 0.05 compared with saline, and the number sign (#) indicates *p* < 0.05 compared with the naive EV group (*n* = 4 per each analysis).

**Figure 8 jcm-08-00533-f008:**
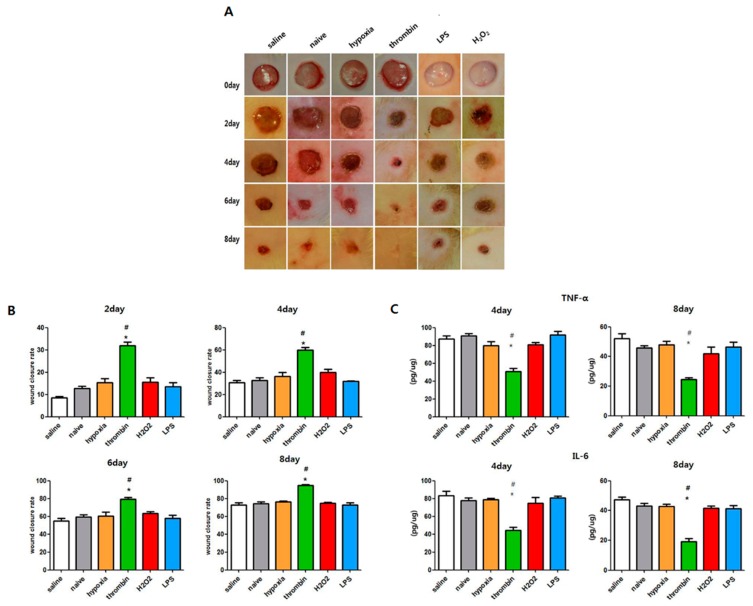
Comparison of wound healing capacity of EVs with the same protein amount (20 μg) in vivo. Punch wounds of animals were treated locally with saline, naive MSC-DERIVED EVs, 10% O2 EVs, thrombin (40 U) EVs, LPS (1 μg) EVs, or H2O2 (50 μM) EVs derived from the same number of MSCs. (**A**) Representative serial wound images in vivo and (**B**) the relative wound closure rate (%) after 2, 4, 6, and 8 days of treatment (**C**) Cytokine analysis for TNF-α and IL-6 β in the homogenate of dissected wound tissues four and eight days after each treatment. Statistical analyses were carried out using a One-way Analysis of Variance (ANOVA). The asterisk (*) indicates *p* < 0.05 compared with saline, and the number sign (#) indicates *p* < 0.05 compared with the naive EV group (*n* = 4 per each analysis).

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
