# Peer review of "Thrombin Preconditioning of Extracellular Vesicles Derived from Mesenchymal Stem Cells Accelerates Cutaneous Wound Healing by Boosting Their Biogenesis and Enriching Cargo Content"

_jcm, 2019, doi:10.3390/jcm8040533_

Reviewer 1 Report

The manuscript from Sung et al. is an in vitro and in vivo study describing the wound healing properties of MSC-derived EVs obtained in different wound mimic preconditioning cell culture media. 

The work is new and interesting, especially in the comparison of the different EV outcomes from the various preconditioning media, and I would find it worthy of publication although there are many points that should be addressed before presenting it to the scientific community.

My general consideration is that English has to be revised before publication especially in the chapters describing the results and the discussion, sometimes it is difficult to understand the meaning of the sentences due to unusual language forms and sentences are too long, I lost track many times there. Furthermore, the authors made a poor work with some of the figures and their legends, which is a pity since the data are interesting. Many graphs lack an adequate description of what has been done, what is the exact scale of the Y-axis and what is measured. 

To help the authors with the revision I listed the most important things to be revised to allow an easier interpretation of the data and to provide all the necessary information:

1)    In the materials and methods there is no paragraph describing in detail the in vivo experimental model, the number of animals used for each condition, the animal welfare, the facility, and the authorization from the pertinent ethics committee, please provide it.

2)    In the legends of most figures there is no brief description of the experiment done, what kind of instrumentation has been used, what is measured and so on, as it is usually done. All the legends should be like the description of figures 5 and 6 which are quite adequate.

3)    Many figures have poor definition and is difficult to read the graphs (Fig 1D, 3C for example).

4)    In legend of fig 1, and also elsewhere, the term “vesicles existed” has to be clarified.

5)    In the legend of fig 1D there is an explanation of the results that should be transferred to the results section (“These particles size distribution displayed about 85% range from 30 to 100 nm.“).

6)    In Fig 2A and B the Y-scale of LPS, H2O2 and Hypoxia conditions should be enlarged to allow a better interpretation of the results.

7)    In the legend of Fig 2C, D and E which cells are used, how much of EVs and incubation time is not described, same for the results section where the results of these experiments are poorly described.

8)    Legend of Fig 3A is not clear, please rephrase.

9)    Title of legend of Fig 7 is not clear, please rename.

10) Fig 7B the Y-axis refers to pg/ml which is not a measure of the wound area extension.

11) Fig. 7D and 8D are not described in the legends.

12)The results section of fig 7 and 8, which should be the core of the paper, are poorly described, in general results should be described more in detail.

13)Please revise the Discussion, sentences are too long, sometimes it is difficult to understand the meaning of the sentences and some concepts are repeated more times.

14)In the Reference chapter, ref 20 is the same as ref 33, ref 34 is just a doi address, please check.

Author Response

Comments and Suggestions for Authors #1

My general consideration is that English has to be revised before publication especially in the chapters describing the results and the discussion, sometimes it is difficult to understand the meaning of the sentences due to unusual language forms and sentences are too long, I lost track many times there. Furthermore, the authors made a poor work with some of the figures and their legends, which is a pity since the data are interesting. Many graphs lack an adequate description of what has been done, what is the exact scale of the Y-axis and what is measured.  To help the authors with the revision I listed the most important things to be revised to allow an easier interpretation of the data and to provide all the necessary information:

à We thank the reviewer for the careful review of our manuscript and the kind comments on our work. We have corrected and improved our manuscript, particularly the figure legends and graphs, in strict accordance with the reviewer’s recommendations. As recommended, we also revised the English language using MDPI English editing service.

In the materials and methods there is no paragraph describing in detail the in vivo experimental model, the number of animals used for each condition, the animal welfare, the facility, and the authorization from the pertinent ethics committee, please provide it.

à In keeping with the reviewer’s recommendation, we have provided paragraphs describing in detail the in vivo experimental model, the number of animals used for each condition, the animal welfare, the facility, and the authorization from the pertinent ethics committee in the Methods section of the revised manuscript (page 8 line 219~239).

 In the legends of most figures there is no brief description of the experiment done, what kind of instrumentation has been used, what is measured and so on, as it is usually done. All the legends should be like the description of figures 5 and 6 which are quite adequate.

à According to the reviewer’s recommendation, we have added a brief description of the experiment, the instrumentation used, and the aspect that was measured. Please check the descriptions of figures 5 and 6 in the revised manuscript. (material and method : page4 line91~97, page 4 line 109~111, page 5 line 125~132, page 6 line 165~169, page 8 line 209-217, page 8 line 220~239; figure legend : page 16 line481~ page18 line 581 )

Many figures have poor definition and is difficult to read the graphs (Fig 1D, 3C for example).

à We have revised the figures and graphs to be more defined, clear, and easier to read. Please see Figures 1D and 3C.

In legend of fig 1, and also elsewhere, the term “vesicles existed” has to be clarified.

à According to the reviewer’s recommendation, we have removed “vesicles existed” in the legend of figure 1 in our revised manuscript. (figure legend : page 16 line 483~491).

In the legend of fig 1D there is an explanation of the results that should be transferred to the results section (“These particles size distribution displayed about 85% range from 30 to 100 nm.“).

à According to the reviewer’s recommendation, we have relocated this sentence to the results section of the revised manuscript. (Result : page 10 line 274~275)

 In Fig 2A and B the Y-scale of LPS, H2O2 and Hypoxia conditions should be enlarged to allow a better interpretation of the results.

à According to the reviewer’s recommendation, we have modified the Y-axis scales in these graphs.(Figure 2, 7 and 8)

In the legend of Fig 2C, D and E which cells are used, how much of EVs and incubation time is not described, same for the results section where the results of these experiments are poorly described.

à According to the reviewer’s recommendation, we have added detailed descriptions of the cells, EV quantity, and incubation time in the legend of Figure 2. (Figure legend : page 16 line 493~503 )

 Legend of Fig 3A is not clear, please rephrase.

à According to the reviewer’s recommendation, we have revised the legend of Figure 3A to ensure that it is clear and detailed.(Figure legend : page 16 line 505~511)

Title of legend of Fig 7 is not clear, please rename.

à  According to the reviewer’s recommendation, we have rewritten title legend of figure 7. (Figure legend : page 18 line 560~567)

Fig 7B the Y-axis refers to pg/ml which is not a measure of the wound area extension.

We thank the reviewer for pointing out the error in Figure 7B, and we apologize for this mistake. We have modified the y-axis statement on the graph(Figure 7)

Fig. 7D and 8D are not described in the legends.

à According to the recommendations by the second reviewer, Figures 7D and 8D should quantitate the characteristics that are claimed as re-epithelialization or collagen formation. We have chosen to remove Figures 7D and 8D. (Figure 7 and 8, Figure legend page 18 line 560~576)

 The results section of fig 7 and 8, which should be the core of the paper, are poorly described, in general results should be described more in detail.

à According to the reviewer’s recommendation, the reviewer’s recommendation, we have rewritten the results sections corresponding to Figures 7 and 8. (Result : page 12 line 371~ page 13 line 385 )

Please revise the Discussion, sentences are too long, sometimes it is difficult to understand the meaning of the sentences and some concepts are repeated more times.

à According to the reviewer’s recommendation, we have removed and rearranged unnecessary sentences. (Discussion : page 13 line 388~ page15 line 463 )

In the Reference chapter, ref 20 is the same as ref 33, ref 34 is just a doi address, please check.

We apologize for our mistake. We have checked the References and ensured that these errors have been removed.(Reference : page 19 line 586~ page 22 line 690)

Reviewer 2 Report

This is an interesting report on maximizing reparative effects of MSC exosomes by pretreating the cells with thrombin.  A concern is that the EV cargo of the thrombin-treated cells contains thrombin, and that the effects noted on cell migration, proliferation and tissue repair are mediated by the thrombin in the EVs. To address this concern the investigators should measure the thrombin content of the EVs and show control experiments that have that amount of thrombin alone added.

Other concerns:

The statement made about MSC (non genetically manipulated) tumorigenicity is not borne out by the literature. References cited do not support that statement, in fact refute it ( reference # 13: “The discrepancies were in part resolved by a recent report by one of the laboratories (12) demonstrating that the transformation of human MSC they observed initially was explained by contamination of their cultures with a small number of malignant cells. “ 

 Please remove this statement from manuscript.

This statement needs modification“Recent studies demonstrated that the beneficial effects of MSCs for wound healing were 53 primarily mediated by secretion of extracellular vesicles.”  Only some are medicated by EV’s. please modify/temper statement.

Please provide some quality control measures of the MSCs used. Purity, multipotency, sterility, karyotype.

Fig 6: scratch assays are generally performed with mitomycin pretreated cells to remove the confounding factor of cell proliferation. Since thrombin is a known mitogen, this approach should be done here.

Figure 7 B legend says that it is the wound area but the axis says pg/ml

There is legend for 7d. In addtion the images should have quantitation of whatever is being claimed as outcome (re-epithelialization, collagen formation) with appropriate N and repeats. If these data are not available, please remove the histological images.

Same for figure 8

All experiments

How often were experiments repeated?

Please don't cite as yet unpublished data,

“We observed best upregulation of protease activated receptor (PAR) 1 and 3 and p-378 ERK 1/2 and p-AKT associated with thrombin preconditioning in our separate study to be reported 379 elsewhere. We are thus conducting further study to delineate the precise role of PAR 1 and 3 and the 380 downstream signaling pathway mediator p-ERK 1/2 and p-AKT for boosting EVs biogenesis and 381 secretion and enriching their cargos.”

Suggest editing by native English language speaker. For example:

“Although the precise molecularmechanisms how preconditioning regimens have not been elucidated yet….”

Author Response

This is an interesting report on maximizing reparative effects of MSC exosomes by pretreating the cells with thrombin.  A concern is that the EV cargo of the thrombin-treated cells contains thrombin, and that the effects noted on cell migration, proliferation and tissue repair are mediated by the thrombin in the EVs. To address this concern the investigators should measure the thrombin content of the EVs and show control experiments that have that amount of thrombin alone added.

à We thank the reviewer for the insightful comments. We also shared the concern that the EV cargo of the thrombin-treated cells might contain thrombin and that the effects noted on cell migration, proliferation, and tissue repair are mediated by the thrombin contained within the EVs. To address this concern, we have already measured the thrombin content in thrombin-treated EVs after isolation and observed no significant differences in the thrombin-treated EVs compared with the negative control and naïve EVs. These perspectives and results were incorporated into the Methods and Results sections of the revised manuscript (Method : page 6 line 165~169 , Result : page11 line 328~330, Figure legend : page18 line 579~581, Supplementary Figure 1)). 

The statement made about MSC (non genetically manipulated) tumorigenicity is not borne out by the literature. References cited do not support that statement, in fact refute it ( reference # 13: “The discrepancies were in part resolved by a recent report by one of the laboratories (12) demonstrating that the transformation of human MSC they observed initially was explained by contamination of their cultures with a small number of malignant cells. “   Please remove this statement from manuscript.

 In keeping with the reviewer’s recommendation, we have removed the statement concerning “MSC tumorigenicity” in the revised manuscript. (Introduction : page 3 line 67~69)

This statement needs modification “Recent studies demonstrated that the beneficial effects of MSCs for wound healing were primarily mediated by secretion of extracellular vesicles.”  Only some are medicated by EV’s. please modify/temper statement. Please provide some quality control measures of the MSCs used. Purity, multipotency, sterility, karyotype.

 à We thank the reviewer for the helpful comments. We have modified this statement in the Introduction section of the revised manuscript (Introduction : page 3 line 70), and we have indicated the quality control measures, purity, multipotency, sterility, and karyotype of the MSCs used in the Methods section of the revised manuscript .(Method : page 4 line 91~ 97)

Fig 6: scratch assays are generally performed with mitomycin pretreated cells to remove the confounding factor of cell proliferation. Since thrombin is a known mitogen, this approach should be done here.

 à We thank the reviewer for the thoughtful comments. In this study, in order to minimize the confounding factor of cell proliferation, cells were incubated into FBS free DMEM medium for 16h instead of mitomycin treatment before EV treatment and scratch assay test. We have revised the method of in vitro scratch-wound-closure assays. (Method : page8 line 209~217)  As the thrombin remnant was not detected in the thrombin preconditioned EVs, it was assumed that the mitogenic effects of residual thrombin in the EVs, if any, might have been minimal in the scratch test of the present study. These views have been incorporated into the revised manuscript (Method : page 6 line 165~169 , Result : page11 line 328~330, Figure legend : page18 line 579~581, Supplementary Figure 1).

Figure 7 B legend says that it is the wound area but the axis says pg/ml

We apologize for our mistake. We have modified the label on the Y-axis in this graph (Figure 7)

There is legend for 7d. In addtion the images should have quantitation of whatever is being claimed as outcome (re-epithelialization, collagen formation) with appropriate N and repeats. If these data are not available, please remove the histological images. 

Same for figure 8

We thank the reviewer for the thoughtful comments. As recommended, we have removed the histological images from Figures 7 and 8 (Figure 7 and Figure 8)

All experiments How often were experiments repeated?

In vitro experiments were repeated three times and in vivo experiments were repeated twice.

Please don't cite as yet unpublished data,  “We observed best upregulation of protease activated receptor (PAR) 1 and 3 and p-378 ERK 1/2 and p-AKT associated with thrombin preconditioning in our separate study to be reported 379 elsewhere. We are thus conducting further study to delineate the precise role of PAR 1 and 3 and the 380 downstream signaling pathway mediator p-ERK 1/2 and p-AKT for boosting EVs biogenesis and 381 secretion and enriching their cargos.”

As recommended by the reviewer, we have deleted the unpublished data in our revised manuscript (Discussion : page14 line 418~424).

Suggest editing by native English language speaker. For example: “Although the precise molecular mechanisms how preconditioning regimens have not been elucidated yet….”

 In accordance with the reviewer’s recommendation, we have revised the grammar and language in English using the English editing service from MDPI Author Service, one of your affiliates. We have also appended the Editing Service Certificate with the revised manuscript.

Round  2

Reviewer 2 Report

my concerns have been adequately addressed.